# Characteristics and Factors Influencing Pore Structure in Shale Oil Reservoirs of Different Lithologies in the Jurassic Lianggaoshan Formation of the Yingshan Gas Field in Central Sichuan Basin

Youzhi Wang [1], Hucheng Deng [2,3,*], Zhiguo Wang [1], Xiandong Wang [1], Qian Cao [3], Dean Cheng [1], Yanping Zhu [1] and An Li [3,*]

1 Exploration and Development Research Institute, Daqing Oilfield Limited Company, Daqing 163712, China
2 State Key Laboratory of Oil and Gas Reservoir Geology and Development Engineering, Chengdu University of Technology, Chengdu 610059, China
3 College of Energy, Chengdu University of Technology, Chengdu 610059, China
* Correspondence: denghucheng@cdut.cn (H.D.); lian@stu.cdut.edu.cn (A.L.); Tel.: +86-198-8115-0998 (H.D.); +86-181-2579-9804 (A.L.)

**Abstract:** Shale in the Jurassic Lianggaoshan Formation in central Sichuan exhibits strong heterogeneity. The study of the pore structure characteristics of different lithologies is crucial to the selection of the target interval. Shale samples of the Lianggaoshan Formation from well YS5 in the central part of the Sichuan Basin were analyzed using scanning electron microscopy, low-temperature nitrogen adsorption, high-pressure mercury injection (HPMI), and large-field splicing method-based scanning electron microscopy (LFS-SEM) to elucidate the pore structure characteristics of shale and their influencing factors. The mineral composition of the reservoir in the study area was diverse, primarily consisting of clay minerals, followed by quartz and calcite. The reservoir space comprised intergranular, granular, and organic matter pores, and oil was observed to fill the reservoir space. Reservoir characteristics varied with the lithological properties. In clayey shale, intergranular pores located in clay mineral particles and pores between pyrite and natural fractures were mainly observed, with a bimodal distribution of pore size and peak distribution of 10–50 nm and >100 nm. The storage space of ash-bearing shale mainly consisted of intragranular pores and intergranular (crystalline) micropores, with pore sizes primarily concentrated in the 10–50 nm range. The storage space in silty shale mainly developed in clastic mineral particles such as quartz, followed by clay mineral intergranular pores with a relatively wide distribution of sizes. Pores were mainly inkbottle-shaped and slit-type/plate-type pores, with an average specific surface area of approximately 6.9046 $m^2 \cdot g^{-1}$ and an average pore volume of approximately 0.0150 $cm^3 \cdot g^{-1}$. The full-pore capillary pressure curve was established using a combination of gas adsorption–desorption tests and HPMI. The fractal dimension of the sample pore structure was calculated, and a significant linear correlation was found between clay mineral content and the fractal dimension. Thus, the pore structure characteristics were mainly controlled by the content and distribution of clay minerals.

**Keywords:** central Sichuan Basin; Lianggaoshan Formation; shale; pore structure characteristics; influencing factors

## 1. Introduction

In recent years, there has been a continuous increase in the exploration and exploitation of shale oil in China, specifically in the Junggar, Ordos, Bohai Bay, Songliao, and Sichuan basins, and shale oil demonstration bases have been established in the Xinjiang, Changqing, Dagang, and Daqing oilfields [1,2]. According to the Ministry of Natural Resources of the People's Republic of China, the geological potential of shale oil resources in China is estimated to be $397.46 \times 10^8$ t, with recoverable resources estimated at $34.98 \times 10^8$ t.

This emphasizes the potential of shale oil as a critical factor for future growth in China's oil reserves [3–6]. The Sichuan Basin has a diverse range of natural gas resources, with the Jurassic strata containing multiple sets of organic black shale that serve as the main interval for oil generation and storage. In 2020, significant breakthroughs were made in the extraction of shale oil [7]. The oil production of the Ping'an 1, Taiye 1, Yuanba 9, and Yuanye 3 wells was 112.8 $m^3/d$, 9.8 $m^3/d$, 16.6 $m^3/d$, and 15.6 $m^3/d$, respectively, and their corresponding gas production amounted to $11.45 \times 10^4$ $m^3/d$, $7.5 \times 10^4$ $m^3/d$, $1.2 \times 10^4$ $m^3/d$, and $1.18 \times 10^4$ $m^3/d$, respectively [8]. After nearly 10 years of development, the technical theories related to shale oil reservoir characteristics have gradually made breakthroughs and improvements. Analysis has evolved from early optical microscopic analysis to current field-emission scanning electron microscopic (FESEM) analysis, low-temperature liquid nitrogen adsorption technology, and the high-pressure mercury intrusion method, among others. It has been upgraded from a single experimental perspective to comprehensive characterization from multiple dimensions. In previous studies, advanced techniques such as scanning electron microscopy backscattering (SEM-BS), SEM, nano-CT scanning, HPMI, and low-temperature $CO_2$ and $N_2$ adsorption experiments have been employed to study the pore structure characteristics of shale reservoirs [9–14]. The results of these studies hold considerable significance for analysis of the pore structure characteristics of shale reservoirs in Jurassic shales of the Sichuan Basin.

However, the abovementioned methods for analyzing the structure of shale oil reservoirs have certain limitations in describing the pore types, morphology, connectivity, and porosity of shale reservoirs. They cannot fully reflect the pore structure characteristics of shale oil reservoirs and cannot achieve quantitative analysis. Continental shale differs from marine shale in that it has strong heterogeneity, low organic matter abundance, low thermal evolution degree, and low hydrocarbon symbiosis and exhibits interbedded distribution of shale, sandstone, and limestone [15,16]. These characteristics make the evaluation criteria of continental shale oil reservoirs challenging to determine, including the selection and prediction of sweet-spot areas [17]. This study utilizes the large-field splicing method based on scanning electron microscopy (LFS-SEM), low-temperature nitrogen adsorption (LTNA), and HPMI to analyze the micropore characteristics of shale reservoirs in the Lianggaoshan Formation of the of the Yingshan gas field. Moreover, using ImageJ software (Fiji, National Institutes of Health, Bethesda, MD, USA) to analyze shale pores, this new image analysis method is more comprehensive and intuitive than identifying pores in a single scanning electron microscopic field of view. It can avoid errors caused by selecting observation positions and can identify various types of pores on a single image, achieving quantitative analysis of pore structure. This study lays the foundation for further research into the pore structure of shale of the Lianggaoshan Formation and its impact on shale oil and gas enrichment [18–21].

## 2. Geological Overview

The Yingshan gas field is located in the central portion of the Sichuan Basin region, spanning three counties: Yingshan, Yilong, and Nanbu. Regionally, the Yingshan gas field belongs to the Yilong tectonic group, which is a low–even belt in the Paleo-Middle Depression of North Sichuan, adjacent to the Paleo-High Middle Oblique Gentle [22,23]. During the early stage of the Himalayan tectonic movement, the central part of the Sichuan Basin underwent north–south extrusion, leading to the development of east–west tectonics in the basin. In the latter stage, the northern part of the study area was subjected to north–south tectonic stress, forming east–west compression structures, as well as the northeast, northwest, and northern knob structural belt tectonics in the central portion of the Sichuan Basin [24].

The Jurassic strata in the study area consist of a series of river–delta–lake formations from the bottom to the top, including the Lianggaoshan Formation and the ZhenzhuChong, Dongyuemiao, Maanshan, and D'Anzhai members of the Zilioujing Formation [25]. The Lianggaoshan Formation represents a typical delta sedimentary system, with receding–strongly

advancing sedimentary cycles [26]. The entire Lianggaoshan Formation consists of mixed-colored shales interspersed with gray siltstones and black shales. The Lower Lianggaoshan Formation, consisting mainly of purple, mixed-colored, and gray–green shales, primarily developed in a floodplain–lakeshore sedimentary environment. The lake basin of the Upper Lianggaoshan Formation underwent three stages: a lake intrusion period, a maximum lake intrusion period, and a lake retreat period [27]. The overall thickness of the upper section is approximately 55–75 m, with the main lithologies being black shale, light gray fine sandstone, siltstone, and gray–black medium shell shale (20–55 m). This is a set of sand and shale rich in organic matter, and it is the primary source and reservoir layer in the study area [28,29]. The upper section of the Lianggaoshan Formation in the study area is primarily composed of shale, sandstone, or tuff with varying thicknesses, with the thicknesses of shale and interbed being approximately 6.36 m and 6.07 m, respectively.

## 3. Materials and Methods

### 3.1. Sample Selection

Eight test samples were selected from the Jurassic Lianggaoshan Formation of well YS5 in the Yingshan area of the Sichuan Basin (Figure 1). The organic matter content of these samples was generally greater than 1.2% and reached up to 3.62%, and the vitrinite reflectance was greater than 0.5%, indicating that they were in the early mature stage for the formation of oil and gas [30–33].

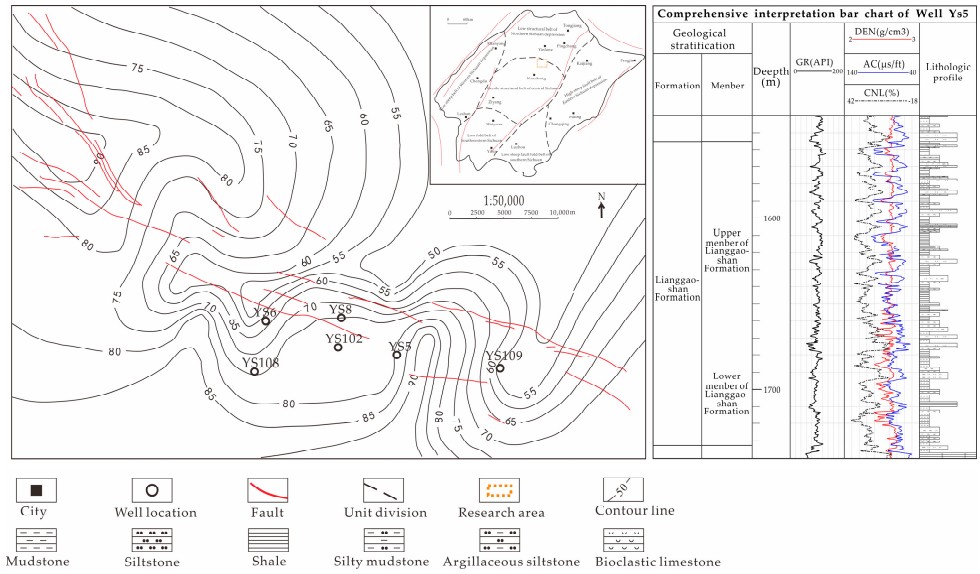

**Figure 1.** Location of the study area and sampling points.

### 3.2. Experimental Methods

The pore structure characteristics of the test samples were analyzed using various methods, including LFS-SEM, gas adsorption ($CO_2$ and $N_2$ adsorption–desorption tests), and HPMI [34]. The distributions of structural characteristics, such as pore size, specific surface area (SSA), and pore volume (PV), of different reservoir space types were classified and statistically analyzed.

LFS-SEM was used to analyze the mineral composition, fluid distribution, and microscopic pore structure characteristics of the test samples by combining the three imaging modes using the field-emission environment. Based on the observation and analysis of the test samples by field-emission SEM, $4 \times 4$, $5 \times 5$, $8 \times 8$, $10 \times 10$, or $10 \times 12$ images were continuously collected from the representative field of view for stitching. Different types of pores were identified by human–computer interaction, and pore sizes and proportions of different storage spaces were classified and counted to achieve qualitative and quantitative analysis (Figure 2).

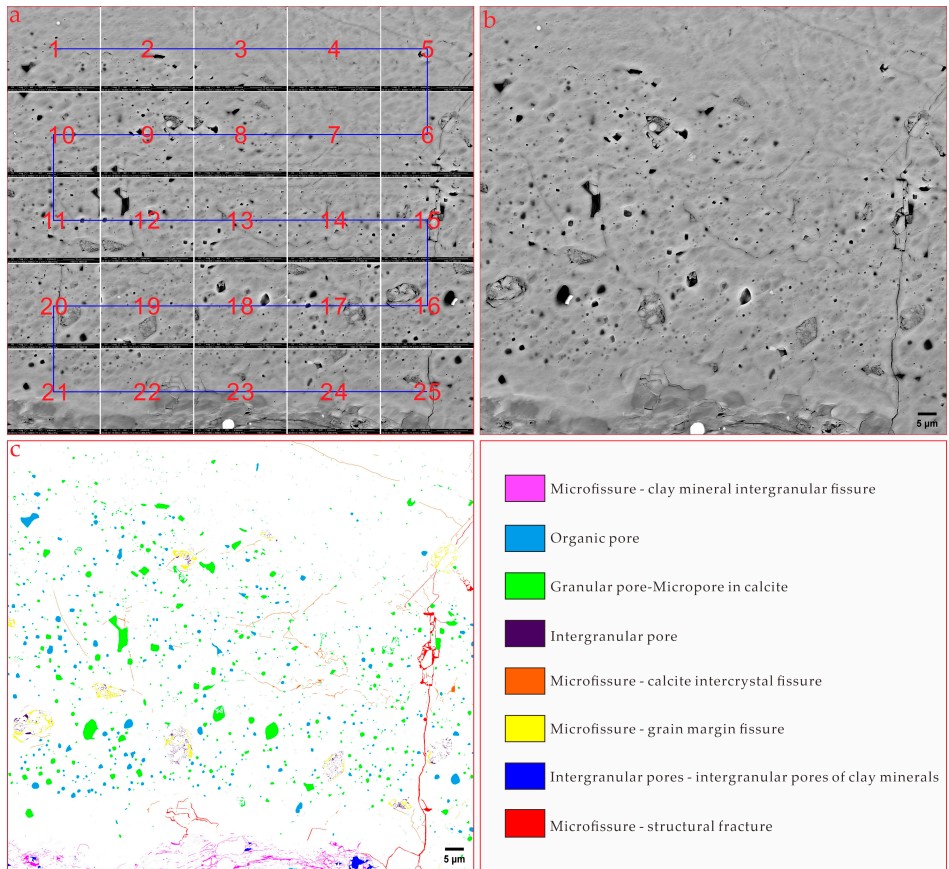

**Figure 2.** Principle and operation process of the large-field-of-view stitching method. (**a**) Splicing of different images by SEM. (**b**) Images observed using SEM at a magnification of 10,000×. (**c**) Identification of different types of storage spaces.

LTNA as conducted using an ASAP2020 adsorption instrument (Quantachrome, Boynton Beach, FL, USA). Based on the static volume method, the determination process of gas adsorption and desorption was completed using the mass conservation equation and static gas balance and pressure, and the adsorption model was preferred to calculate the specific surface, PV, and pore diameters.

HPMI was conducted using a Poremaster-60 instrument (Quantachrome, Boynton Beach, FL, USA). In HPMI, liquid mercury is gradually injected into the pores of test samples under a certain external pressure. The mercury is a non-wetting phase that is injected to replace the wetted phase, and additional pressure is required to inject this non-wetting-phase liquid into the rock pores. The amount of mercury injected provides the volume of the pores and is controlled by the pore throat sizes. The maximum mercury injection pressure was approximately 418 MPa. The capillary pressure curve and pore distribution curve of the shale sample were obtained by varying the injection pressure.

The X-ray diffraction (XRD) test steps are as follows: the sample is washed to a fluorescence level below 4, then dried at a temperature below 60 °C until the sample is cooled to room temperature, ground until all particles are less than 40μm in size, and finally put into the groove of the sample slide for measurement. After the measurement, the X-ray diffraction pattern is obtained, according to which the relevant data are read, then compared with the standard X-ray diffraction data of minerals to determine the type of minerals, and the contents of various minerals and clays are calculated using the following formula.

$$X_i = \left[ \frac{\frac{I_i}{K_i}}{\sum \frac{I_i}{K_i}} \right] * 100$$

where:

X$_i$ is he percentage content of mineral i, expressed as a percentage;
I$_i$ is the diffraction peak intensity of mineral i; and
K$_i$ is the reference strength of mineral i.

## 4. Characteristics of the Reservoir Space of the Lianggaoshan Formation

### 4.1. Lithological Characteristics

The reservoir of the Lianggaoshan Formation in the study area is characterized by rocks with diverse mineral compositions, mixed particle sizes, and complex rock structures, as observed through cores, thin sections, and SEM. Clastic particles, mainly composed of quartz, debris, and feldspar, are scattered throughout the rocks, with grain sizes predominantly in the range of 0.03–0.06 mm. Sand rich layers are locally developed, with small amounts of organic matter distributed in the mud as clumps and carbonate particles. such as calcite distributed in the mud as grains, partly filling the intergranular spaces and accounting for the edges of the clastic particles. The lithology of the Lianggaoshan Formation primarily comprises clay shale, sand shale, and laminated blocky gray shale (Figure 3). X-ray diffraction (XRD) reveals that clay minerals dominate the mineral fraction of the shale in well 5 in the study area, followed by quartz, calcite, plagioclase, and pyrite. The clay mineral content ranges from 41.40% to 58.10%, with an average content of 47.28%, whereas quartz content ranges from 28.9% to 39.70%, with an average content of 33.60%. Calcite content varies from 1.70% to 21.90%, with an average content of 9.97%. The predominant clay minerals in the shale are illite, with an average content of approximately 38.00%, followed by kaolinite, chlorite, and illite/smectite (Table 1).

Combined with core observation, thin section identification, scanning electron microscopic observation, and mineral composition determination results, the test samples of the Lianggaoshan Formation mainly include clay shale, silty shale, and lime-bearing shale. The clay shale has a uniform distribution of mud-grade debris and undeveloped laminae. The silty shale lamina is developed, the lamina shape is relatively straight, and the lamina type is mainly a felsic silt thin layer. Mud crystal dolomite and powder crystal dolomite are widely developed in the layered massive limestone-bearing shale. The composition is mainly calcite, with a portion of dolomite. A small amount of clay minerals is distributed between calcite particles. Detritus limestone and debris laminae are developed, and the composition is mainly calcite.

**Table 1.** Basic information about test samples in the upper section of the study area.

| Number | Depth (m) | TOC (%) | Mineral Content (%) | | | | | | Clay Mineral Content (%) | | | |
|---|---|---|---|---|---|---|---|---|---|---|---|---|
| | | | Quartz | Feldspar | Calcite | Dolomite | Pyrite | Clay Mineral | Illite | Kaolinite | Chlorite | Illite/ Montmorillonite Mixed Layers |
| 2 | 1611.44–1611.55 | 3.17 | 32.9 | 4.5 | 3.9 | 0 | 0.6 | 58.1 | 23.2 | 9.3 | 17.4 | 8.1 |
| 4 | 1612.57–1612.71 | 3.62 | 28.9 | 2.4 | 19.5 | 1.3 | 3.6 | 44.3 | 14.2 | 10.2 | 11.5 | 8.4 |
| 6 | 1614.69–1614.86 | 1.35 | 30.3 | 2.3 | 19.8 | 1 | 3.1 | 43.5 | 17.0 | 12.2 | 9.1 | 5.2 |
| 8 | 1615.64–1615.79 | 0.64 | 28.9 | 2 | 21.9 | 2.2 | 3.6 | 41.4 | 14.9 | 11.2 | 9.9 | 5.4 |
| 10 | 1647.04–1647.20 | 1.64 | 34.4 | 7.8 | 13 | 0 | 2.4 | 42.4 | 18.7 | 5.5 | 12.7 | 5.5 |
| 12 | 1648.75–1648.96 | 1.22 | 35.2 | 9.2 | 0 | 0 | 2 | 53.6 | 24.1 | 7.0 | 14.5 | 8.0 |
| 14 | 1652.54–1652.67 | 1.46 | 39.7 | 9.1 | 0 | 0 | 1.3 | 49.9 | 19.0 | 5.0 | 9.5 | 16.5 |
| 16 | 1653.09–1653.27 | 1.08 | 38.5 | 12 | 1.7 | 2.8 | 0 | 45 | 13.5 | 6.3 | 9.0 | 16.2 |

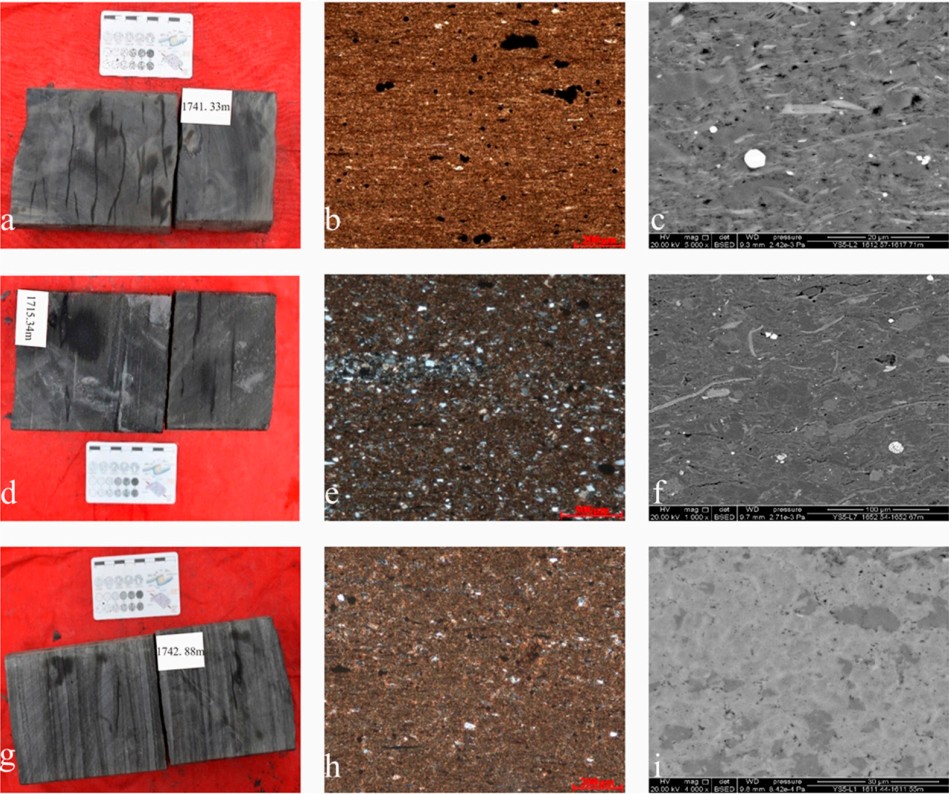

**Figure 3.** Rock cores, thin sections, and scanning electron microscopic images of the reservoir rocks of the Lianggaoshan Formation. (**a**) Clay shale; 1741.88 m. (**b**) Clay shale with organic matter distributed sporadically in agglomerates; 1658.50 m, 10 × 10(−), scale bae = 200 μm. (**c**) Clay shale with high clay mineral content and intergranular micropores; 1612.57 m, 5000×. (**d**) Siltstone; 1715.34 m. (**e**) Sand shale, quartz, and other mineral grains are dispersed, with visible as streaks; 1714.54 m, 5 × 10(+), scale bae = 500 μm. (**f**) Sand shale showing long, ingrained grains (grain size 10–30 μm); 1652.54 m, 1000×. (**g**) Gray shale showing ash-textured laminae; 1742.58 m. (**h**) Gray shale showing calcite granules distributed in mud; 1717.54 m, 10 × 10(+), scale bae = 200 μm. (**i**) Gray shale showing grainy calcite laminae with intracrystalline pores; 1611.44 m, 4000×.

*4.2. Storage Space Type and Distribution Characteristics*

The exploration of unconventional oil and gas resources involves the study of reservoir space type, porosity, permeability, and fracture characteristics. These factors are crucial in the process of oil and gas accumulation [35]. Shale oil and gas reservoirs exhibit complex pore structures and diverse space types that serve as both reservoir fluid spaces and flow spaces [36–39]. Several researchers worldwide have proposed various classification schemes for pore structure types and characteristics of shale reservoirs. These schemes classify the pore types into four categories based on factors such as mineral structure, organic matter content, and diagenesis. The pore types include intragranular pores, intergranular pores, organic matter pores, and microfractures. By utilizing SEM and argon ion polishing, the shale storage space of the Lianggaoshan Formation in the study area can be classified into two categories: pores and microfractures. The pores comprise intergranular pores, intragranular pores, organic matter pores, and intergranular microfractures of clay minerals. Moreover, oil stains can be observed filling the reservoir space.

The intergranular pores are mainly distributed between clay or detrital particles, and a large number of intergranular pores were retained during the compaction of clay mineral particles and detrital particles under the action of sedimentation and diagenesis. The intergranular pores are irregularly shaped (e.g., polygons or sheets Figure 4a–d), and they were mainly distributed between different crystals formed by accumulation during the regrowth of different mineral crystals. Samples from the Lianggaoshan Formation are

rich in pyrite, calcite, and other carbonate minerals. Calcite exhibited a microcrystalline structure, and pyrite was spherical and showed a cluster-like distribution. Microscopic observation of the intergranular pores in samples from the Lianggaoshan Formation mainly revealed calcite intergranular pores, pyrite intergranular pores, and some clay mineral intergranular pores (Figure 4e–g). These types of pores were irregularly distributed, and the morphology was related to the crystal morphology and accumulation method. The pores were small in size.

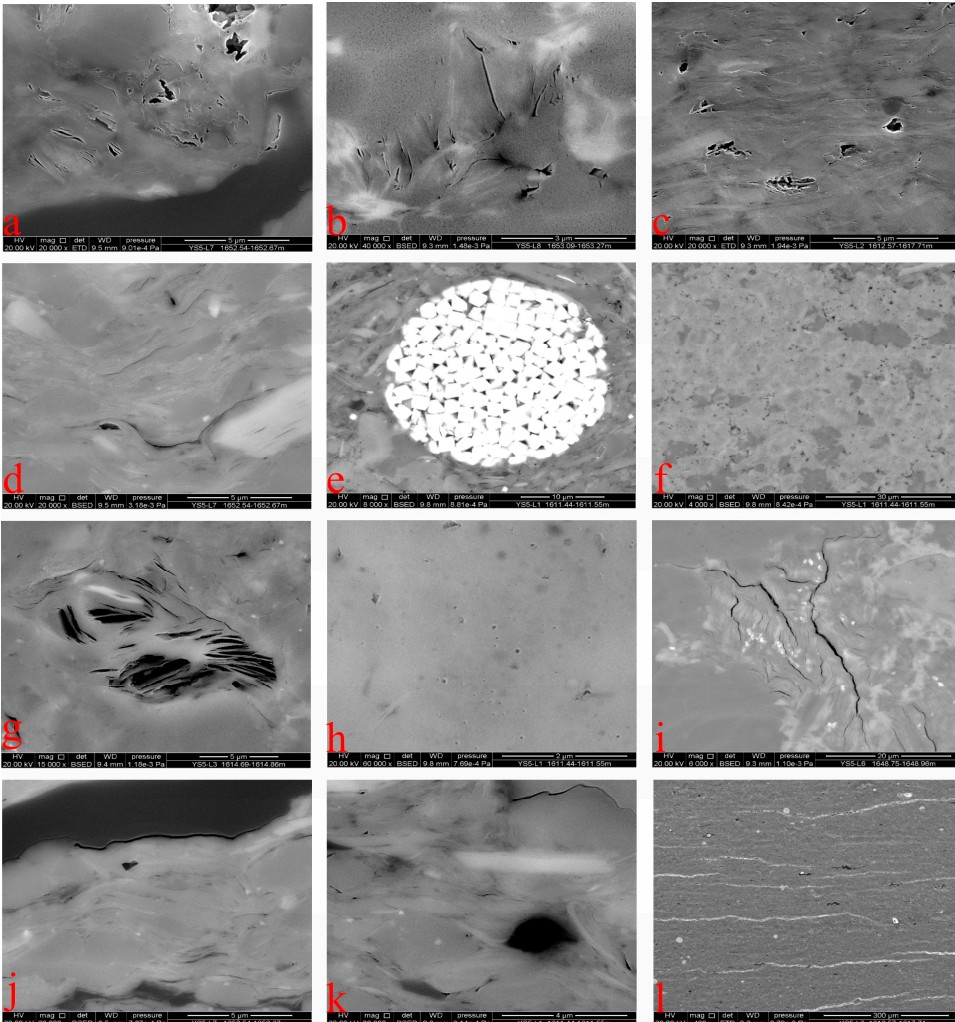

**Figure 4.** Spatial distribution of different types of shale reservoirs in the study area. (**a**) Intergranular pore development between clay minerals; Yingshan 5-L7, 1652.54 m. (**b**) Intergranular pore development between clay minerals; Yingshan 5-L8, 1653.09 m. (**c**) Intergranular pore development between oil-filled clay minerals; Yingshan 5-L2, 1612.57 m. (**d**) Intergranular pore development between linearly distributed clay minerals; Yingshan 5-L7, 1652.54 m. (**e**) Intercrystalline pores developed between globular pyrite grains; Yingshan 5-L1, 1611.44 m. (**f**) Intercrystalline pores of calcite; Yingshan 5-L1, 1611.44 m. (**g**) Intercrystalline pore development between clay minerals; Yingshan 5-L3, 1614.69 m. (**h**) Intracrystalline solution pores; Yingshan 5-L1, 1611.44 m. (**i**) Intercrystalline microslits of clay minerals; Yingshan 5-L6, 1648.75 m. (**j**) Organic matter with contraction joints developed; Yingshan 5-L7, 1652.54 m. (**k**) Contraction joints in a quartz mineral grain; Yingshan 5-L1, 1611.44 m. (**l**) Laminated joints; Yingshan 5-L2, 1612.57 m.

The intragranular pores were mainly distributed in quartz, clay minerals, and calcite grains. Among these, the content of calcite in the samples from the Lianggaoshan Formation was high, and the pores in the granules were mainly dissolution pores that formed when

calcite was dissolved by acidic fluids (Figure 4h). The intragranular pores were independent and irregularly developed, and the pore size was mainly distributed in the orders of nano- and micro-nanometers. The development of microfractures can effectively improve the deep flow capacity of the reservoir. The microfractures in the shales of the Lianggaoshan Formation were mainly laminae, grain edge joints, or organic matter contraction joints, among which the grain edge joints were prominent. These grain edge joints were primarily distributed in a mesh pattern with good interconnectivity. The laminae joints with large pore sizes were mainly distributed among different clay minerals or mineral crystals, such as rhodochrosite (Figure 4i–l). The development of reservoir space types in different lithology samples is quite different.

Observations with a magnification range of 200–20,000× and statistical analysis of all pore types using LFS-SEM revealed variations in the distribution of characteristic parameters such as the number and size of different types of pores. The proportion of different pore types indicated that intergranular pores and microfractures were more prominent, accounting for 49.93% and 25.51% of the total number of pores, respectively. This was followed by organic matter pores and intragranular pores, accounting for 15.62% and 9.44% of the total number of pores, respectively. The sizes of pores ranged from 10 nm to 10 μm, mainly comprising nanoscale pores (≤1 μm). The sizes of intergranular pores varied across a wide range, with the peak pore size mainly distributed in the range of 0.1–1 μm, whereas the sizes of organic matter pores varied across a relatively small range, with the peak pore size being <0.01μm. The sizes of microcracks were observed to range from a few nanometers to a few micrometers (Figure 5).

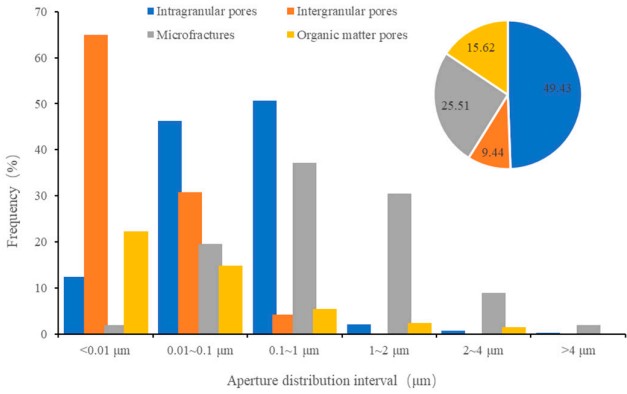

**Figure 5.** Distribution frequency of different pore types and sizes.

### 4.3. Pore Structure Distribution Characteristics

To comprehensively characterize the pore structure and the distribution of different pore sizes in the shale of the Lianggaoshan Formation in the central part of the Sichuan Basin, various experimental methods were used in this study. Gas adsorption–desorption data of the samples ($CO_2$ and $N_2$) were interpreted using the multipoint BET model and the BJH model, and the samples were characterized in terms of SSA, pore size, and PV. The connectivity of the test samples was evaluated based on the results of HPMI tests. Furthermore, full-pore capillary pressure curves were established by combining LTNA and HPMI tests, and the fractal dimension of the pore structure of different samples was calculated. The fractal dimension of the pore structure was calculated to quantitatively characterize the structural features of the reservoir space distribution across the full pore size range of the samples.

The pore diameter distribution of clay shale and its pore characteristics play a critical role in the evaluation of shale oil reservoirs. There are several pore diameter classification schemes for shale reservoirs, and the International Union of Pure and Applied Chemistry (IUPAC) classification, which defines micropores as pores with diameters between 0 and 2 nm, mesopores as pores with diameters between 2 and 50 nm, and macropores as pores with diameters greater than 50 nm, is currently used. This classification scheme was utilized

in the present study. Eight samples from the Lianggaoshan Formation in the study area were subjected to LTNA, and the results indicated that the $CO_2$ adsorption curves of the samples were similar to the type I isothermal adsorption line, as defined by the IUPAC. When the relative pressure (P/Po) was less than 0.01, the $CO_2$ adsorption increased as P/Po increased, and when P/Po was greater than 0.01, $CO_2$ adsorption increased gradually as P/Po increased (Figure 6a), suggesting that the test sample had developed a certain number of micropores and that the filling of these micropores had occurred. The isothermal adsorption–desorption curve displayed an inverse "S" shape as P/Po increased, which is a type II isotherm (Figure 6). The adsorption–desorption curves indicated that both the adsorption and desorption of shale steadily increased when the relative pressure was low, and the desorption curve had a large inflection point when P/Po was in the range of approximately 0.4~0.5, consistent with the inflection point calculated by Harris et al. [40–42]. When P/Po was close to 1, the adsorption volume started to increase rapidly, the adsorption and desorption curves tended to overlap, and the hysteresis loop gradually closed.

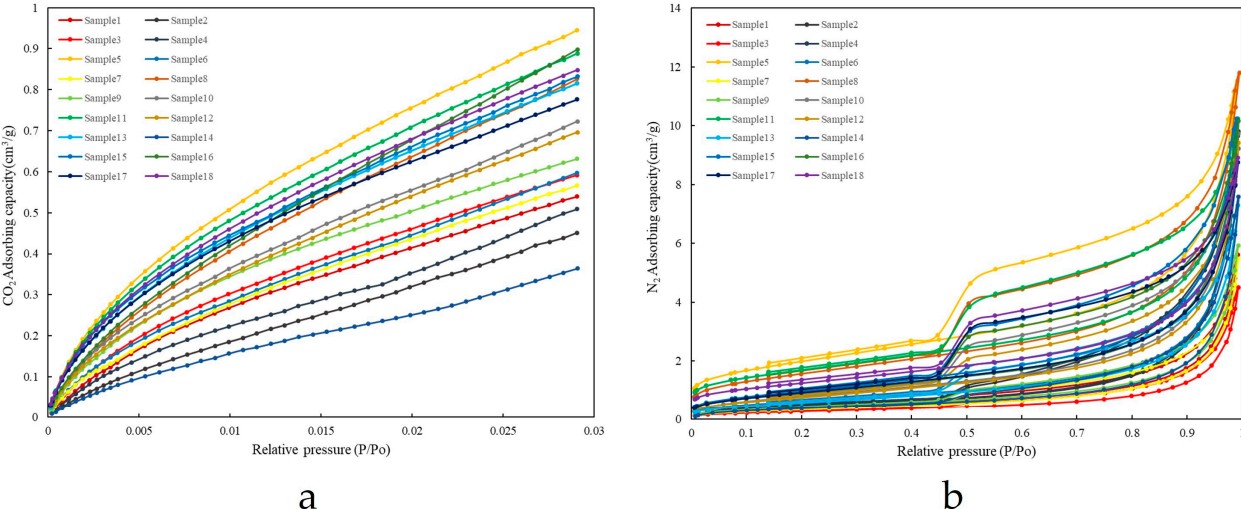

**Figure 6.** Gas adsorption–desorption curves of shale samples from the Lianggaoshan Formation in the study area. (**a**) $CO_2$ gas adsorption–desorption curve. (**b**) $N_2$ gas adsorption–desorption curve.

The pores in the tested samples were categorized into two groups based on their pore structure and whether a hysteresis loop could be generated: class I isothermal adsorption curves were of H2–H3 type, and in the samples at 1614.69 m, 1615.64 m, 1647.04 m, 1648.75 m, and 1653.09 m, the corresponding pores were mainly open pores, including parallel-walled slit-type capillary pores with open edges and cylindrical pores with open ends. Class II isothermal adsorption curves were of H3–H4 type, and in the samples at 1611.44 m, 1612.57 m, and 1652.54 m, the corresponding pores were mainly ink bottle pores with thin necks and wide bodies, which were closed at one end but could produce a hysteresis return line; there was a sharp decreasing inflection point in the desorption curve. The characterization results are consistent with the above SEM observation results.

The gas adsorption and desorption data were analyzed using the multipoint BET model and the BJH model to determine the specific surface area (SSA) and pore volume (PV) of the samples. The SSA values of the test samples ranged from 3.9061 to 11.1376 $m^2 \cdot g^{-1}$, with an average of 6.9046 $m^2 \cdot g^{-1}$, and PV values ranged from 0.0117 to 0.0185 $cm^3 \cdot g^{-1}$, with an average of 0.0150 $cm^3 \; g^{-1}$. The SSA and PV values of the samples corresponding to the class I isothermal adsorption curves were large, and micropores were developed. Clay shale has the largest specific surface area, with an average of about 9.13 $m^2 \cdot g^{-1}$, while lime-bearing shale has the smallest specific surface area, mainly distributed at 7.27 $m^2 \cdot g^{-1}$.

The morphology of the mercury pressure curve is indicative of the development of pore space in each pore throat section and the connectivity between pore spaces. Based on

Figure 7, the capillary pressure curve, as a whole, is observed to be skewed towards the upper right, with the central main inlet mercury section being flat and long. To analyze the pore characteristics of the samples, the discharge drive pressure, sorting coefficient, maximum mercury inlet saturation, and mercury withdrawal efficiency were all considered. The pore structures in the study area were broadly categorized into two major types. The first type (Figure 7) included the samples taken at depths of 1611.44 m and 1614.69 m, where the mercury inlet curve exhibited a stepwise increase as the pressure rose. The curve could be divided into two sections, with the displacement pressure being less than 1.0 MPa, corresponding to a maximum pore throat radius of approximately 10 μm. The samples in this category had a relatively large maximum mercury injected saturation (up to 70%) and a mercury withdrawal efficiency of approximately 50%. The morphology of these pores was primarily ink-bottle-like, with micropores that were generally undeveloped, with small pore throats and poor connectivity. For the second type (Figure 7, including the samples taken at depths of 1612.57 m, 1615.64 m, and 1653.09 m), the mercury inlet curve displayed a sudden increase with increasing pressure, and the gentle section of the mercury inlet curve was relatively short. The sorting coefficient was small, and the pore distribution was relatively uniform. The displacement pressure was relatively high, at approximately 4.0 MPa, and the maximum mercury inlet saturation of the samples was small, ranging from 40% to 50%. The PV was relatively small, and the maximum mercury rejection efficiency reached 70%, indicating that the pores in the samples were predominantly slit-type or flat-type micropores with good connectivity.

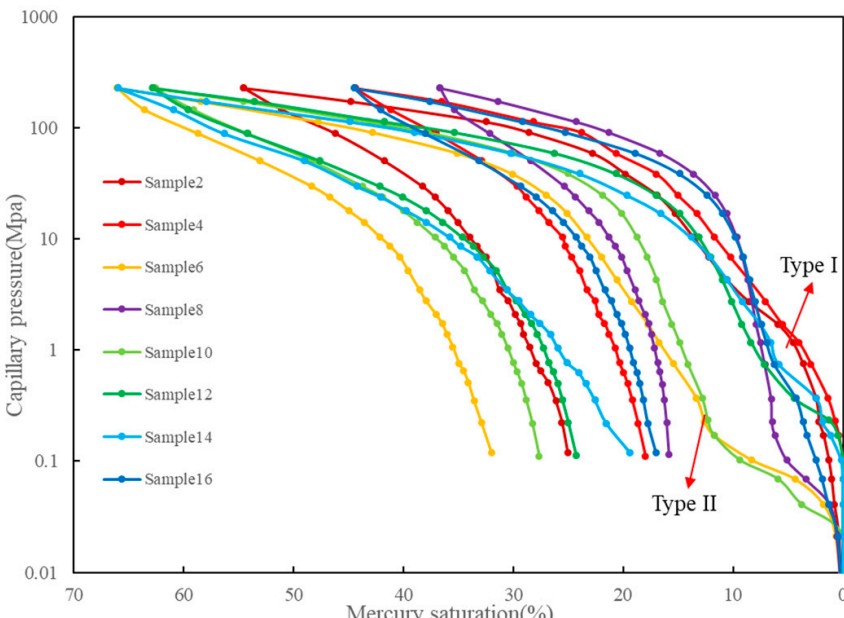

**Figure 7.** Mercury pressure curves of samples from the Lianggaoshan Formation in well YS5.

The full-scale pore size distributions of various samples were computed based on energy industry standard NB/T 14008-2015 of the People's Republic of China. The results indicated that the full-scale storage capacity of the samples was primarily governed by mesopore (2–50 nm) distributions. The peak values were mainly observed in the range of 20–30 nm. Additionally, some micrometer-scale macropores were observed in the sand shale samples (Figure 8).

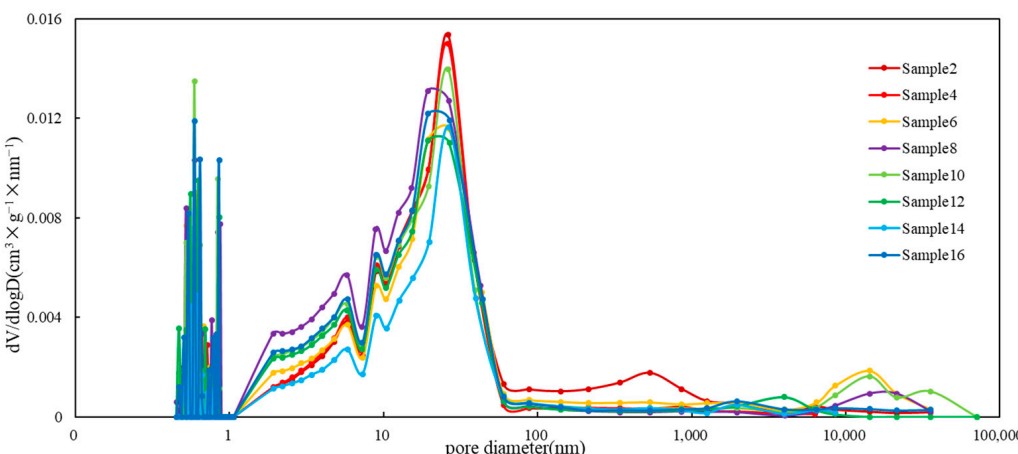

**Figure 8.** Pore size distribution curve of test samples from the Lianggaoshan Formation in Well YS5 in the central portion of the Sichuan Basin.

## 5. Factors Influencing the Development of the Pore Structure

### 5.1. Comparison of Pore Structure Characteristics of Different Lithologies

Using LFS-SEM, the pore size distribution of various lithologies was quantitatively analyzed, revealing notable differences in the pore size amongst different lithologies. As demonstrated in Figure 5, samples 1 and 7 are clay shales that feature a bimodal pore size distribution, with the peaks predominantly distributed in the ranges of 10–50 nm and >100 nm, respectively. Furthermore, samples 6 and 13 are gray shales with a pore size distribution in the range of 10–200 nm, mainly concentrated in the range of 10–50 nm. Samples 2 and 14 correspond to sand shales that exhibit a relatively broad pore size distribution, mainly in the range of 10–2000 nm. The pore size distribution of these samples, particularly that of samples 2 and 14, is relatively broad, with micrometer- and millimeter-sized pores primarily distributed in the range of 10–2000 nm, mainly concentrated in the range of 40–100 nm (Figure 5). Combined with SEM results, it can be seen that a certain number of micropores and microfractures are present in the clay shale samples, primarily consisting of intergranular pores of clay minerals, intergranular pores of pyrite, and natural fractures, indicating good overall pore connectivity. The pore size distribution in the gray shale is quite broad, with micropores mainly consisting of calcite intergrain pores and mineral intragranular pores, and occasionally featuring paleontological intracavity pores.

### 5.2. Factors Influencing Pore Structure Development

The pore structure characteristics of shale are affected by several factors. The sedimentary environment influences the content of organic carbon, brittle minerals, and clay minerals within shale. The SSA and PV of shale are influenced by the content of organic matter, clay minerals (e.g., illite), and brittle minerals (e.g., authigenic quartz). Not only is the total organic carbon content an important indicator of source rock quality, but it also has a considerable impact on the PV, SSA, and adsorbed gas content of the reservoir, particularly when comparing shale in the high maturity and overmaturity stages. The degree to which intergranular pores, intragranular pores, and microfractures develop is influenced by the content of brittle minerals, which is an important criterion used to determine whether shale oil and gas has commercial exploitation value. The type and content of clay minerals influence the conversion of clay minerals during diagenetic evolution and thereby influence the PV and SSA of micropores within them. The samples examined in this study were primarily fresh samples collected from the same drilling, and the depths at which the samples were obtained did not vary significantly. Thus, organic matter maturity did not have a significant influence on the pore structure, and the effects of organic carbon content, clay mineral content, and brittle mineral content (e.g., quartz and feldspar) on the characteristics of pore structure in different lithologies were mainly studied in this work [43]. The pore

fractal dimension, a quantitative measurement of pore structure complexity and surface roughness, exhibited a highly consistent change trend with that of reservoir porosity, SSA, and PV. Based on the results of LTNA and HMPI, a full pore size capillary pressure curve was developed to analyze the full pore size distribution characteristics of shales from the Lianggaoshan Formation in the study area [44]. The fractal dimension of the pore structure within the full pore size range was then calculated using the following equation:

$$lgS = (3 - D)lgR - (3 - D)lgR_{max}$$

where R is the pore size corresponding to the capillary pressure (Pc) in the reservoir (μm); S is the saturation of the wetted phase in the reservoir when the capillary pressure is Pc (%); and D is the fractal dimension, which is a dimensionless quantity.

The fractal dimension values of shale samples were calculated (Table 2), and the correlation between the fractal dimension and organic carbon mass lithology, clay minerals, brittle minerals (e.g., quartz and feldspar), and carbonate mineral content, was analyzed in this study (Figure 9). The correlation between brittle minerals, such as felsic minerals, and the fractal dimension was not very clear. Quartz and feldspar possess a regular crystal structure and form, their intergranular and intragranular pores are not well-developed, their morphology is relatively singular, and their SSA and PV are relatively limited. Thus, these factors had a relatively small effect on the fractal dimension of the pore structure. In contrast, a significant linear correlation was observed between clay mineral content and the fractal dimension, indicating that the pore structure characteristics of the shale in the Lianggaoshan Formation were primarily controlled by clay minerals [45].

**Table 2.** Distribution of SSA, PV, and the fractal dimension of test samples from the Lianggaoshan Formation.

| Number | SSA (m$^2$/g) | PV (cm$^3$/g) | Fractal Dimension of Full Aperture Range |
|---|---|---|---|
| 1 | 4.2003 | 0.008694 | |
| 2 | 5.2561 | 0.014928 | 2.7843 |
| 3 | 2.8221 | 0.006958 | |
| 4 | 5.3439 | 0.015185 | 2.7005 |
| 5 | 16.0593 | 0.017826 | |
| 6 | 7.1863 | 0.014279 | 2.7299 |
| 7 | 3.7634 | 0.008465 | |
| 8 | 14.4304 | 0.018242 | 2.7124 |
| 9 | 4.5694 | 0.009177 | |
| 10 | 9.575 | 0.015047 | 2.7568 |
| 11 | 13.7266 | 0.015825 | |
| 12 | 9.27 | 0.014558 | 2.7535 |
| 13 | 6.3728 | 0.011286 | |
| 14 | 4.1966 | 0.011727 | 2.7209 |
| 15 | 10.3870 | 0.015830 | |
| 16 | 10.3587 | 0.015705 | 2.7514 |
| 17 | 9.7045 | 0.013542 | |
| 18 | 10.6579 | 0.013794 | |

The clay minerals in the samples were primarily illite and kaolinite. Illite is a 2:1 type phyllosilicate mineral, while kaolinite is a typical 1:1 type phyllosilicate mineral. Their unique crystal structures give clay minerals porosity and large surface areas. Under SEM, illite often appears as flaky or filamentous, with rectangular, wedge-shaped, or slit interlayer pores often formed between parallel illite plates. As the clay mineral content increases, the development of clay mineral granular (crystalline) pores and microfractures becomes the main contributor to storage space in the samples. Furthermore, organic matter, which can take the form of complexes between clay mineral particles, can also coexist with clay minerals in pores (Figure 3b,e). The interaction between clay minerals and organic matter promotes the thermal evolution of organic matter, leading to the formation of a certain

amount of organic matter micropores, thereby increasing the fractal dimension of pore distribution. The heterogeneity of pore distribution is strong. Carbonate minerals, intergranular dissolution pores, and intragranular dissolution pores are relatively developed, and their shapes are mostly circular or elliptical. This type of pore distribution is relatively uniform, and the fractal dimension of pore distribution decreases(Figure 10), indicating that the presence of carbonate minerals such as calcite contributes to the development of pores to a certain extent.

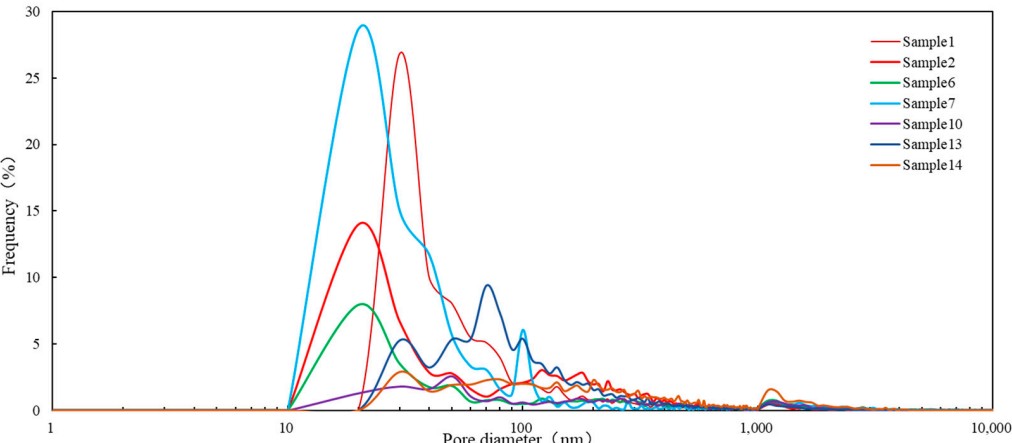

**Figure 9.** Pore size distribution in samples of different lithologies of the Lianggaoshan Formation in the central part of the Sichuan Basin.

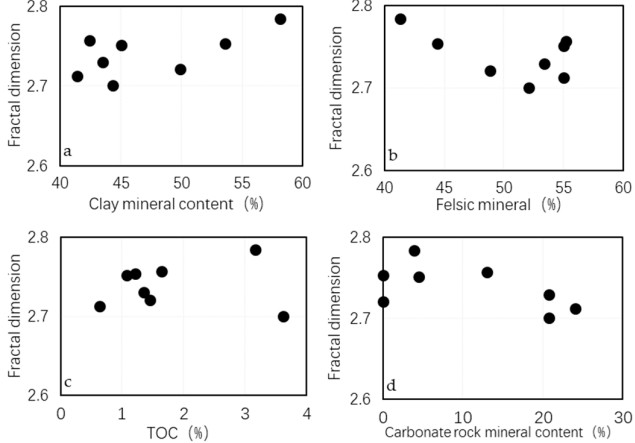

**Figure 10.** Fractal dimension of TOC and major mineral compositions versus full pore size range. (**a**) Relationship between clay minerals content and fractal dimension. (**b**) Relationship between Felsic mineral and fractal dimension. (**c**) Relationship between TOC and fractal dimension. (**d**) Relationship between Carbonate rock mineral content and fractal dimension.

## 6. Conclusions

*Recommendations for Hydraulic Fracturing Modifications based on In Situ Stress Magnitude Evaluation*

(1) The reservoir of the Lianggaoshan Formation in the study area exhibits a diverse rock mineral composition and a complex rock structure. XRD analysis, SEM, and energy spectrum tests reveal that the lithology of the Lianggaoshan Formation primarily consists of clay shale, sand shale, and gray shale. The shale reservoir space of the Lianggaoshan Formation in the study area can be classified into two categories: micropores and microfractures. The pores consist of intergranular pores, intragranular pores, and organic matter pores, and traces of oil filling in the reservoir space are observed. The intergranular pores and microfractures are relatively well-developed (accounting for 49.93% and 25.51% of the total

number of pores, respectively), followed by organic matter pores and intragranular pores. The predominant pore morphologies are ink-bottle-type, slit-type, or plate-type, with the average SSA being approximately 6.9046 $m^2 \cdot g^{-1}$ and the average PV being approximately 0.0150 $mL \cdot g^{-1}$. The storage space is primarily composed of mesopores with a distribution range of 2 to 50 nm, with the peaks mainly located between 20 and 30 nm.

(2) The variations in pore structure characteristics among different lithologies are considerable. In clay shale, a considerable number of micropores and microfractures are observed, which are primarily intergranular pores, interpyrite granular pores, and natural fractures, exhibiting good pore connectivity overall. Gray shale exhibits intragranular pores and intergranular (crystalline) micropores as the primary storage space, with pore sizes ranging from 10 to 50 nm. In sand shale, the reservoir space is primarily developed between clastic mineral particles, such as quartz, with microfractures occurring between clay minerals. The content of clay minerals in various lithologies exhibits a substantial linear correlation with the fractal dimension. As the clay mineral content increases, the quantity of intergranular pores and intergranular microfractures within the samples increases, indicating that clay minerals primarily control the pore structure characteristics.

**Author Contributions:** Conceptualization, Y.W. and H.D.; writing—original draft preparation, Q.C.; writing—review and editing, A.L.; supervision, Z.W.; resources, X.W.; data curation, D.C.; investigation, Y.Z. All authors have read and agreed to the published version of the manuscript.

**Funding:** This research was supported by National Natural Science Foundation of China (No. 42072182), and Sichuan Outstanding Young Scientific and Technological Talents Project (No. 2020JDJQ0058).

**Institutional Review Board Statement:** Not applicable.

**Informed Consent Statement:** Not applicable.

**Data Availability Statement:** Not applicable.

**Acknowledgments:** The authors would like to thank all the reviewers who participated in the review.

**Conflicts of Interest:** The authors declare no conflict of interest.

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
