# Peer review of "Characteristics and Factors Influencing Pore Structure in Shale Oil Reservoirs of Different Lithologies in the Jurassic Lianggaoshan Formation of the Yingshan Gas Field in Central Sichuan Basin"

_minerals, doi:10.3390/min13070958_

Round 1
Reviewer 1 Report
The authors employed multiple approaches to investigate the pore structure characteristics of shale samples from the Lianggaoshan formation. This work is of interests to this journal. However, it would benefit from some revision before it can be considered for publication. Comments and suggestions for improving the quality of the manuscript are given as follows:
1. The introduction should provide an overview of the relevant research progress, identify existing defects and problems to be addressed in this paper, as well as highlight the innovative contributions of this study.
2. Line 108, ‘I The pore structure…’. Please check it.
3. Line 140, ‘A The reservoir of the Lianggaoshan …’. Please check it.
4. Overall, this manuscript resembles a report more than an academic paper. It is necessary to clarify the innovation and contribution of this research, as well as its guiding significance for practical applications.
Moderate editing of English language is necessary.
Author Response
请参阅附件

Reviewer 2 Report
Dear editor,
I’m now ready to conclude my reviews on the manuscript Characteristics and Factors Influencing Pore Structure in Shale Oil Reservoirs of Different Lithologies in The Jurassic Lianggaoshan Formation of The Yingshan Gas Field in Central Sichuan Basin from Wang et al., submitted to Minerals. I would like to congratulate the authors that provided a robust manuscript integrating different methods; that is what we expect in a good manuscript. The manuscript provides a detailed analysis of the pore structure characteristics of shale samples from the Lianggaoshan Formation in the Sichuan Basin. The study utilizes various analytical techniques, such as SEM, HPMI, ... , to investigate the mineral composition and pore structure of the shale samples. The introduction provides relevant background information on the significance of shale oil exploration in China and the potential of shale oil resources in the Sichuan Basin. The geological settings satisfactorily describe the Yingshan gas field in the Sichuan Basin. The materials and methods are clear and provide a comprehensive view of the methods utilized, and following this, the results are clearly presented and well illustrated.
Overall, the text provides a comprehensive analysis of the pore structure characteristics of shale samples from the Lianggaoshan Formation, and this is a good study. It effectively presents the research methodology, experimental results, and their interpretation, including figures and statistical data, enhancing the clarity and understanding of the findings. However, the main question for me is whether the text could benefit from further discussion and interpretation of the results, particularly concerning their implications for shale oil and gas exploration in the Sichuan Basin and other areas. What novelty does this study bring?. I Believe including this in the manuscript could increase the audience and impact of the manuscript; if this kind of discussion is not applied, it is a study case. For sure, it also could be made to include a few more references to reinforce the author's main conclusions. Finally, again I expect the authors to explore the implications of their results for the reservoir deeply.
Additionally, Some minor grammatical errors and typos are present, which are in the pdf attached.
Based on this, I would like to recommend
With the best regards

Some minor grammatical errors and typos are present, which are in the pdf attached.
Author Response
请参阅附件。

Reviewer 3 Report
Methods applied by the authors are correct but there are only 18 point samples. It is not acceptable from the statistical point of view. If Authors revieved several hundred or thousends tresults of analysis they must show the way of choice this 18 .
If not additional analyses are absolutely needed.
Round 2
Reviewer 1 Report
The majority of issues have been addressed, but the paper's innovation requires further enhancement.
The language requires refinement by professional institutions or native English speakers.
Author Response
Please refer to the attachment for reply comments.

Reviewer 2 Report
Dear Editor,
After revising the revised version of the manuscript I would like to recommend accepting the manuscript in its present form.
I would like to congratulate the authors of the manuscript on the very well and clear manuscript. The text was well-written and clear.
With best regards.
There are a few minor typos.
e.g., line 307 (. the)
Again, the text was well-written and clear.
Author Response

(The authors gave the same response as above.)

Reviewer 3 Report
Only comments have changed not experimental facts. Regional characterization on the base of 18 samples it is nonsence
Author Response

(The authors gave the same response as above.)
